# Abnormal Hypermethylation of CpG Dinucleotides in Promoter Regions of Matrix Metalloproteinases Genes in Breast Cancer and its Relation to Epigenomic Subtypes and HER2 Overexpression

**DOI:** 10.3390/biomedicines8050116

**Published:** 2020-05-10

**Authors:** Olga A. Simonova, Ekaterina B. Kuznetsova, Alexander S. Tanas, Viktoria V. Rudenko, Elena V. Poddubskaya, Tatiana V. Kekeeva, Ivan D. Trotsenko, Sergey S. Larin, Sergei I. Kutsev, Dmitry V. Zaletaev, Marina V. Nemtsova, Vladimir V. Strelnikov

**Affiliations:** 1Molecular Genetic Diagnostics Laboratory 2, Research Centre for Medical Genetics, Moskvorechie St 1, 115522 Moscow, Russia; simonova_o.a@mail.ru (O.A.S.); shkarupo@mail.ru (V.V.R.); 2Epigenetics Laboratory, Research Centre for Medical Genetics, Moskvorechie St 1, 115522 Moscow, Russia; kuznetsova.k@bk.ru (E.B.K.); tanas80@gmail.com (A.S.T.); kekeeva@mail.ru (T.V.K.); kutsev@mail.ru (S.I.K.); zalnem@mail.ru (D.V.Z.); nemtsova_m_v@mail.ru (M.V.N.); 3Medical Genetics Laboratory, I.M. Sechenov First Moscow State Medical University (Sechenov University), Trubetskaya St 8-2, 119991 Moscow, Russia; 4Clinic of Personalized Medicine, I.M. Sechenov First Moscow State Medical University (Sechenov University), Trubetskaya St 8-2, 119991 Moscow, Russia; podd-elena@yandex.ru; 5VitaMed LLC, Seslavinskaya St 10, 121309 Moscow, Russia; 6Institute of Medicine, RUDN University, Miklukho-Maklaya St 6, 117198 Moscow, Russia; trotsenkoivan@mail.ru; 7Molecular Immunology Laboratory, Federal Scientific Clinical Centre of Pediatric Hematology Oncology Immunology Named after Dmitry Rogachev, Samory Mashela St 1, 117997 Moscow, Russia; sergei_larin@mail.ru; 8Gene Therapy Laboratory, Institute of Gene Biology, Vavilova St 34/5, 119334 Moscow, Russia

**Keywords:** matrix metalloproteinases, tissue inhibitors of matrix metalloproteinases, DNA methylation, breast cancer, methylation-sensitive restriction enzyme digestion PCR (MSRE-PCR), genome-wide DNA methylation analysis

## Abstract

Matrix metalloproteinases (MMPs) and their tissue inhibitors (TIMPs) substantially contribute to the regulation of intercellular interactions and thereby play a role in maintaining the tissue structure and function. We examined methylation of a subset of 5’-cytosine-phosphate-guanine-3’ (CpG) dinucleotides in promoter regions of the *MMP2, MMP11, MMP14, MMP15, MMP16, MMP17, MMP21, MMP23B, MMP24, MMP25*, *MMP28,*
*TIMP1, TIMP2, TIMP3*, and *TIMP4* genes by methylation-sensitive restriction enzyme digestion PCR. In our collection of 183 breast cancer samples, abnormal hypermethylation was observed for CpGs in *MMP2, MMP23B, MMP24, MMP25*, and *MMP28* promoter regions. The non-methylated status of the examined CpGs in promoter regions of *MMP2, MMP23B, MMP24, MMP25*, and *MMP28* in tumors was associated with low HER2 expression, while the group of samples with abnormal hypermethylation of at least two of these MMP genes was significantly enriched with HER2-positive tumors. Abnormal methylation of *MMP24* and *MMP25* was significantly associated with a CpG island hypermethylated breast cancer subtype discovered by genome-wide DNA bisulfite sequencing. Our results indicate that abnormal hypermethylation of at least several MMP genes promoters is a secondary event not directly functional in breast cancer (BC) pathogenesis. We suggest that it is elevated and/or ectopic expression, rather than methylation-driven silencing, that might link MMPs to tumorigenesis.

## 1. Introduction

Matrix metalloproteinases (MMPs) are a broad group of extracellular proteinases that specifically hydrolyze the extracellular matrix proteins. MMPs are involved in various processes, such as cell matrix development and remodeling; cell adhesion, migration, differentiation, and proliferation; tissue repair; angiogenesis; embryo development [1]. MMPs are inhibited by a group of tissue inhibitors of MMPs (TIMPs). The family is currently known to include four members: TIMP1, TIMP2, TIMP3, and TIMP4 [2].

The roles that MMPs and TIMPs play in tissue development and function are the subject of intense research, and new data continuously become available in the field. To date, both protein families have been implicated in rheumatoid arthritis, osteoarthritis, parodontitis, autoimmune lesions of the skin, and instability of atherosclerotic plaques [3,4]. Data on the role of MMPs in cancer deserve special attention. MMPs are capable of modeling the tumor microenvironment by degrading the extracellular matrix and affecting signal transduction by interacting with growth factors [5]. MMPs can play a role in invasion, angiogenesis, the formation of premetastatic tumor niches, and antitumor immunity [6,7]. Changes in the expression levels of members of the MMP and TIMP families have been described in various cancers, including breast cancer (BC) [8].

MMPs are involved in a number of processes associated with the development of the breast carcinoma. Their interactions with TGFβ, EGFR, and Fas modulate proliferation and apoptosis of tumor cells [9]. MMPs are able to promote invasion of transformed cells due to the degradation of extracellular matrix molecules. In addition, a number of MMPs are involved in tumor invasion by disrupting cell–cell contacts and activating cell motility and migration [10,11]. Many MMPs can directly activate epithelial–mesenchymal transition (EMT). As an example, MMP3 directly promotes EMT by activating alternative splice pathways which result in the expression of Rac1b activated splice variant with unique tumor-promoting activities, which stimulates epithelial–mesenchymal transition by increasing levels of cellular reactive oxygen species [12].

The contribution of MMPs and TIMPs to cancer development is intriguing to study. In this work, the methylation status of the promoter regions was assessed for genes of the two families in order to understand the mechanisms of their epigenetic regulation in normal tissues and to identify new DNA methylation markers in breast cancer.

## 2. Materials and Methods

### 2.1. Clinical Material

We examined 183 BC samples and 183 matched samples of morphologically normal adjacent tissues obtained prior to chemotherapy, five BC cell lines (ZR711, HS578T, BT474, T47D, and MCF7), and six autopsy samples of normal breast tissues. Biological material was obtained from the Blokhin Russian Cancer Research Center, Gertsen Moscow Research Cancer Institute, Research Centre for Medical Genetics, Institute of Gene Biology, and Russian Scientific Center of Roentgenoradiology. Altogether, 378 samples were analyzed in this study. All BC and matched samples were obtained at surgery of cancer cases none of which underwent neoadjuvant chemotherapy. All tissue samples were fresh-frozen. Fragments of tissues for DNA analysis were examined macro- and micromorphologically: expression of ER, PR, and HER2 receptors was evaluated by immunohistochemistry. The approximate amount of tumor cells in each sample was estimated to exceed 80%. Fragments of about 10 mg were used for DNA extraction.

This study was approved by the Institutional Ethics Committee of the Research Centre for Medical Genetics. Written informed consent was obtained from each participant of this study.

The median age of BC patients and its standard deviation were 57 ± 11.56 years.

Data on the histological type were available for 176 samples. Of these, 75.5% (133/176) were identified as ductal BC; 12% (21/176) as lobular BC; 8% (14/176) as mixed BC. Micropapillary carcinoma and mucinous, medullary, metaplastic, and low-differentiated tumors were diagnosed in single cases.

Data on the disease stage and TNM classification were available for 178 patients. According to the TNM classification, the sample collection was stratified by the size of the primary tumor (T1, 20.2% (36/178); T2, 68.5% (122/178); T3, 5.6% (10/178); T4, 5.6% (10/178)); the status of regional lymph nodes (N0, 46.6% (83/178); N1, 44.4% (79/178); N2, 8.4% (15/178); N3, 0.6% (1/178)); and distant metastasis (M0, 97.8% (174/178) and M1, 2.2% (4/178)).

### 2.2. DNA Isolation and Methylation-Sensitive Restriction Enzyme Digestion PCR (MSRE-PCR)

Genomic DNA was isolated by standard phenol–chloroform extraction. The DNA digestion mixture for the methylation-sensitive restriction enzyme digestion contained 1.5 μg of genomic DNA, 10 units of HpaII restriction endonuclease, and 2 μL of a SEBufferY buffer (×10) (SibEnzyme, Novosibirsk, Russia). Deionized water was added to bring the final volume to 20 μL, and the mixture was incubated at 37 °C for 16 h.

### 2.3. Methylation-Sensitive Restriction Enzyme Digestion PCR (MSRE-PCR) Assays

Triplex MSRE-PCR assays were used for each locus under study, where one fragment was amplified from the target gene, another one served as a positive PCR control (a constitutively methylated region of the *CUX1* gene [13]), and a third one was used to check the completeness of DNA hydrolysis (a constitutively non-methylated region of *SNRK* [13]). The nucleotide sequences of the primers are shown in Table 1.

PCR reactions were performed as described earlier [13]. The MSRE-PCR products were resolved by electrophoresis in 8% polyacrylamide gel and stained with silver nitrate (Figure 1).

### 2.4. Bisulfite Sequencing by Sanger

The results of the analysis of promoter methylation of target genes obtained by MSRE-PCR were verified with bisulfite sequencing of corresponding fragments. To perform bisulfite conversion, genomic DNA was denatured in NaOH (at a final concentration of 0.3 M) at 65 °C for 15 min. DNA was modified using sodium bisulfite and hydroquinone taken at final concentrations 2 and 0.5 M, respectively, for 15 h at 55 °C. Modified DNA was purified using Wizard DNA Cleanup system (Promega, Madison, Wisconsin, USA) according to the manufacturer’s instructions. PCR reactions were performed as described earlier [13]. PCR products were sequenced with an ABI3100 genetic analyzer using terminating dideoxynucleotides according to the protocol for ABI Prism 3100 Genetic Analyzer (Thermo Fisher Scientific, Waltham, Massachusetts, USA). The representative bisulfite sequence is shown in Figure 2.

### 2.5. Validation of MSRE-PCR Results by RRBS

For the validation of MSRE-PCR results by RRBS, two RRBS datasets were used, one from the ENCODE project [14], and another from our previous XmaI-RRBS study [15] performed on a subset of 111 BC samples and six normal breast samples from the collection described here. XmaI-RRBS was performed as described by us earlier [16]. A representative example is shown in Figure 3.

## 3. Results

We performed selective screening of methylation of 5’-cytosine-phosphate-guanine-3’ dinucleotides (CpGs) located in the promoter regions of MMPs and TIMPs genes in DNA extracted from 183 surgical fresh-frozen samples of BC and matched morphologically normal breast tissues, and six autopsy samples of normal breast tissues, as well as from five BC cell lines, by methylation-sensitive restriction enzyme digestion PCR (MSRE-PCR).

Methylation analysis of the selected CpG dinucleotides located in the promoter regions of the MMPs and TIMPs genes by MSRE-PCR has divided the genes into three categories:
Non-methylated in normal breast tissues but prone to abnormal hypermethylation in breast cancer (*MMP2, MMP23B, MMP24, MMP25*, and *MMP28*; with the proportion of methylated samples indicated in Table 1)Non-methylated in both normal breast tissues and in breast cancer (*TIMP2, TIMP3, MMP11, MMP15, MMP16*, and *MMP17*)Constitutively methylated in normal and cancerous breast tissues (*TIMP1, TIMP4, MMP14*, and *MMP21*)

MSRE-PCR results were selectively validated by Sanger bisulfite sequencing of the same samples and by comparing MSRE-PCR results for the same samples with two RRBS datasets, one from the ENCODE project [14], and another from our previous XmaI-RRBS study [15] performed on a subset of 111 BC samples and six normal breast samples from the collection described here.

Breast cancer cell lines were analyzed alongside with the clinical samples in order to validate the results of our MSRE-PCR assays, and to provide reference information that can be reevaluated by other researchers. By now, the ENCODE project [14] collection of data on DNA methylation obtained by reduced representation bisulfite sequencing (RRBS) contains information regarding two of the cell lines assessed in our study, MCF7 and T47D. For these, our MSRE-PCR results demonstrated in Table 2 exactly recapitulate the ENCODE RRBS data, approving the validity of our approach. Altogether, non-methylated status of the promoter CpG islands of all the genes from the first two categories shown above is in line with the ENCODE RRBS results for the normal breast tissue sample (BC_Breast_02-03015; breast, donor 02-03015, age 21, Caucasian, DNA extract).

Further analysis was focused on the matrix metalloproteinases genes in which we have identified CpGs that are non-methylated in normal breast tissues but are prone to abnormal hypermethylation in breast cancer.

A multiple correspondence analysis was performed with the data on the promoter CpGs methylation of the genes under study. The most intriguing results were obtained when the HER2 expression status was tested for association with the methylation of the MMPs genes (Figure 1). The non-methylated status of the CpGs under study in promoter regions of *MMP2, MMP23B, MMP24, MMP25*, and *MMP28* in tumors appeared to be associated with lack of HER2 expression. In addition, the methylated status of the *MMP23B* promoter CpGs was associated with a highest level (3+) of HER2 expression (Figure 4).

A cluster analysis was carried out on the methylation data of *MMP2, MMP23B, MMP24, MMP25,* and *MMP28* differentially methylated promoter CpG dinucleotides (Figure 5). A remarkable cluster of samples was enriched in MMP genes methylation, in which CpGs of at least two MMP genes are concurrently methylated in each sample. This cluster appeared to be significantly enriched in HER2-positive (HER2 ihc scores 2 and 3) tumors (*p* = 0.012). With HER2 hic score threshold drawn between score 0 and scores 1–3 the differences were less obvious and did not reach statistical significance (*p* = 0.09), which we discuss below.

Correlations between MMP genes abnormal methylation and other molecular or clinical features of tumors (ER, PR status; TNM; grade) were not statistically significant.

We have previously performed methylotyping of 100 BC samples from the collection described here using the genome-wide CpG islands bisulfite DNA sequencing (XmaI-RRBS, XmaI-reduced representation bisulfite sequencing), and demonstrated that the samples segregate into two major subtypes, highly and moderately methylated [15]. In order to assess if abnormal methylation of the MMP genes is merely a reflection of the attribution of the samples to the highly methylated subtype, we aligned the XmaI-RRBS data onto the samples track provided by unsupervised clusterization of the MSRE-PCR DNA methylation data obtained for the *MMP2, MMP23B, MMP24, MMP25,* and *MMP28* genes in this study. Based on the results shown in Figure 5, we hypothesize that abnormal methylation of *MMP24* and *MMP25* might suggest a hypermethylated subtype for the tumor, while abnormal methylation of *MMP23B* is most likely independent of the global methylation subtype. Indeed, Fisher’s exact test supports a statistically significant association of abnormal methylation of *MMP24* and *MMP25* promoters in tumor samples with their attribution to the hypermethylated BC subtype with *p* = 0.033 for *MMP24* and *p* = 0.002 for *MMP25*.

## 4. Discussion

A balance of synthesis and degradation, as well as the proper functioning of the components of the extracellular matrix is the fundamental basis of tissue morphogenesis. Matrix metalloproteinases play a key role in this process. Collectively, they are able to degrade a wide spectrum of extracellular matrix proteins [17]. Considering the fact that, in addition to structural components, an enormous number of proteins with diverse functions (including signaling molecules) circulate in the extracellular space, MMPs are able to influence a vast range of aspects of cell life. By participating in the extracellular field remodeling, MMPs play an extraordinary role in a variety of cellular processes, such as cell adhesion and migration, signal transduction, cell differentiation, and proliferation [18].

In this study, we used methylation-sensitive restriction enzyme digestion PCR (MSRE-PCR) as a principal method to assess DNA methylation of CpG dinucleotides in the promoter regions of 15 target genes in a collection of 378 samples. MSRE-PCR is a very simple and affordable approach to CpG methylation assessment requiring small amounts of DNA, which makes it attractive to use when collections of significant sample size are under study. Yet, MSRE-PCR imposes certain limitations on the results, because only a small subset of methylation sites in the promoter regions of the genes can be addressed, namely the ones that harbor a recognition sequence for the restriction enzyme used. Methylation in these positions does not fully reflect the promoter methylation despite the general trend of DNA methylation being correlated within a certain proximity. As in our study we used DNA methylation for the purpose of screening for differentially methylated CpGs rather than for comprehensive characterization of DNA methylation along the whole length of the gene promoter sequences, we consider the MSRE-PCR method appropriate. The validity of the results obtained in our study is also supported by comparison with RRBS datasets performed either on the same samples or on biologically related samples (normal breast tissue and exhaustively studied BC cell lines).

As MSRE-PCR is a qualitative rather than a quantitative method, we did not perform laser microdissection of the tissue samples prior to DNA extraction. At the same time, MSRE-PCR is very sensitive to methylated alleles. It can identify methylation in a heterogeneous mix containing <2% of cells with methylated fragments [19]. This allows us to use MSRE-PCR for abnormal DNA methylation screening in clinical samples for which a fraction of tumor cells is only roughly estimated.

By MSRE-PCR, we identified abnormal hypermethylation of promoter CpG dinucleotides of five MMP genes, *MMP2, MMP23B, MMP24, MMP25,* and *MMP28*, in breast cancer. By now, the most comprehensive dataset on DNA methylation in BC is provided by The Cancer Genome Atlas (TCGA). We gained access to TCGA data through an interactive viewer Wanderer [20], to explore MMPs DNA methylation in breast cancer, with level 3 TCGA data for methylation arrays (450k Infinium chip). It should be borne in mind, that 450k Infinium chip probes do not cover each CpG dinucleotide in the gene promoter CpG islands. TCGA data for the probes that coincide with, or are in close vicinity to, the loci assessed by MSRE-PCR in our study, well support hypermethylation of these regions of the *MMP2, MMP23B, MMP24, MMP25,* and *MMP28* promoters in BC, providing additional validation to our data.

In our study, abnormal hypermethylation of promoter CpG dinucleotides of the *MMP2, MMP23B, MMP24, MMP25,* and *MMP28* was associated with HER2-positive tumor status, and, to a different extent, with CpG island hypermethylated epigenomic BC subtype. HER2-associated abnormal hypermethylation of CpG dinucleotides in gene promoters is to be interpreted in light of previously reported results also obtained for breast cancer samples.

In 2009, Terada et al. reported an association between increased number of methylated genes and HER2 amplification in breast cancer [21]. They found out that the incidence of HER2 amplification was significantly higher in the cancers with frequent methylation than in those with no methylation, which completely coincides with our results. Moreover, the number of methylated genes correlated with the degree of HER2 amplification in their study. In our study, we also detected gradual increase in association between genes methylation and HER2 amplification when choosing a higher HER2 score as a threshold. Overall, our results together with those published earlier [21], although obtained on completely different sets of genes, suggest that frequent methylation has a strong association with HER2 amplification in breast carcinomas. Yet, as both our study and that of Terada et al. [21] were performed on limited sets of specially selected genes, it is intriguing to investigate what these sets have in common that would explain similar results obtained in our experiments. Terada et al. selected genes whose silencing is unlikely to confer growth advantage and avoided selection bias of cells with methylation. To this end, they selected genes primarily based on the absence of expression in normal human mammary epithelial cells [21]. In our study we selected for DNA methylation analysis MMP and TIMP genes that possessed CpG islands in their promoters assuming the possibility of the regulatory role of abnormal methylation of these genomic regions in cancer. Overall, we analyzed 15 selected genes and identified five that demonstrate abnormal methylation in breast cancer, as well as association of abnormal methylation with HER2-positive status and hypermethylated epigenetic BC subtype. Surprising as it may seem, what brings these MMP genes together with those studied by Terada et al. [21], is that they are not or are negligibly expressed in the normal breast, and most are specifically expressed in other organs: *MMP23B* in arteries and ovary, *MMP24* in cerebellum, *MMP25* in whole blood, *MMP28* in tibial nerve and testis (the gene expression data were obtained from the GTEx Portal https://www.gtexportal.org on 04/15/20).

Thus, the MMP genes that have demonstrated abnormal methylation of their promoters in HER2-positive breast tumors have no potential to be downregulated by methylation in BC, as they are not expressed in normal breast. What is then the reason, the mechanism and the clinical significance of *MMP2, MMP23B, MMP24, MMP25,* and *MMP28* genes promoter methylation in HER2-positive BC?

Activated HER2 receptors activate the downstream signaling through multiple pathways, including PI3K/AKT/mTOR, resulting in induction of cell cycle progression [22]. Altered DNA methylation is thought to be an early event in BC and the number of genes affected was reported to increase with progression [23]. It was proposed that epigenetic changes like methylation may also affect key players in the PI3K/AKT/mTOR pathway [24]. Our results, on the contrary, suggest that increase in abnormal CpG island methylation driven by HER2 is mediated by PI3K/AKT/mTOR pathway leading to enhanced proliferation and rapid tumor evolution with accumulation of CpG island hypermethylation that is not directly functional in BC pathogenesis.

Experimental research [25,26,27,28], together with the results described here, suggest that it is rather elevated and/or ectopic expression, than methylation-driven silencing, that might link MMPs to tumorigenesis. As for the MMPs analyzed here, it was shown for MMP23B that it promotes cell invasiveness on MDA-MB−231 breast cancer cells [26]. Hyperexpression of MMP24 was described in BC [27]. A study with the SKOV(3) ovarian cancer cell line has shown that MMP24 may facilitate cancer cell invasion [28]. Higher levels of the *MMP25* mRNA in tumor tissues have been observed in astrocytoma, glioma, rectal cancer, and prostate cancer [29]. Functional studies with MMP25-overexpressing colorectal cancer cell lines (HCT−116 and HT−29) have shown that MMP25 upregulation correlates with an increased tumor growth after subcutaneous cell grafting in mice [30].

## 5. Conclusions

In this study, we set out to analyze abnormal DNA hypermethylation at promoter CpG islands of matrix metalloproteinases and their tissue inhibitors genes in breast cancer, suggesting that such hypermethylation might lead to gene inactivation and subsequently disrupt concerted regulation of tumor microenvironment, thus promoting cancer progression and metastases. We identified CpG dinucleotides prone to abnormal hypermethylation in BC in five MMP genes, *MMP2, MMP23B, MMP24, MMP25,* and *MMP28,* but failed to detect any association of their hypermethylation with clinical or molecular features of the tumors, except for the associations with HER2-positive phenotype and CpG island hypermethylated epigenetic subtype of BC. The common characteristic of these MMP genes, sufficiently explaining absence of clinical/molecular correlations, is their absent or negligible expression in normal breast. Thus, their promoter methylation is most likely a passenger epigenetic mutation obtained and kept in the rapidly evolving HER2-positive tumors, where activated HER2 receptors induce accelerated cell cycle progression. This suggestion is supported by previously published results obtained in HER2-positive BC samples for a completely different gene set [21].

We here demonstrate that, though hypermethylation of certain CpGs in promoter regions of the *MMP2, MMP23B, MMP24, MMP25,* and *MMP28* genes is cancer specific, it lacks independent biological or clinical value, being rather a function from HER2 activation. Generally, such considerations should always be borne in mind in clinical epigenetic research. Finally, we suggest, that for MMP genes it is rather elevated and/or ectopic expression, than methylation-driven silencing, that might link them to tumorigenesis.

## Figures and Tables

**Figure 1 biomedicines-08-00116-f001:**
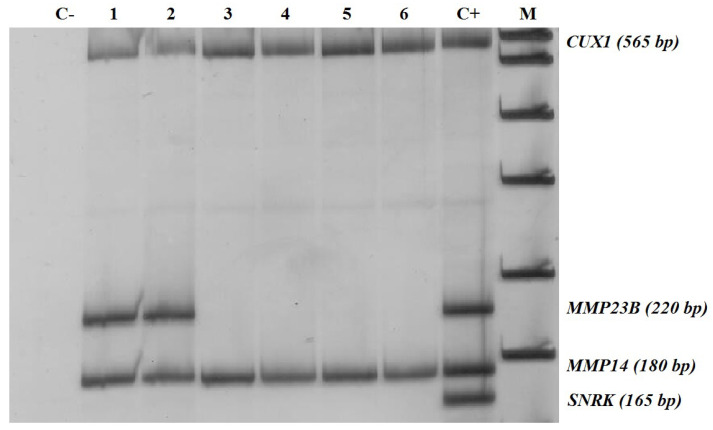
*MMP14* and *MMP23B* simultaneous analysis by methylation-sensitive restriction enzyme digestion PCR (MSRE-PCR). M, DNA ladder pUC19/HpaII; K+, MSRE-PCR products obtained with an undigested human genomic DNA as a template; 1–6, MSRE-PCR products obtained with breast cancer genomic DNA samples digested with HpaII. Positions of the PCR products corresponding to the *MMP14* and *MMP23B* promoter 5′-cytosine-phosphate-guanine-3′ (CpG) islands under analysis, as well as a positive PCR control (a constitutively methylated region of the *CUX1* gene), and a DNA digestion control (a constitutively nonmethylated region of *SNRK*) are specified on the right. MSRE-PCR products from *MMP14* are detectable in all the samples, corresponding to constitutively methylated status of its promoter CpG island in all normal and cancerous breast tissues. *MMP23B* bands are visible only in a K+ sample and in samples 1–2, and are absent in samples 3–6, reflecting differentially methylated status of this fragment. MSRE-PCR does not provide information on the methylation status of individual CpGs contained within the restriction enzyme recognition sequence in an assessed locus. Thus, positive MSRE-PCR signal was interpreted as hypermethylation of the whole target locus, while negative MSRE-PCR signal, as its non-methylated state.

**Figure 2 biomedicines-08-00116-f002:**
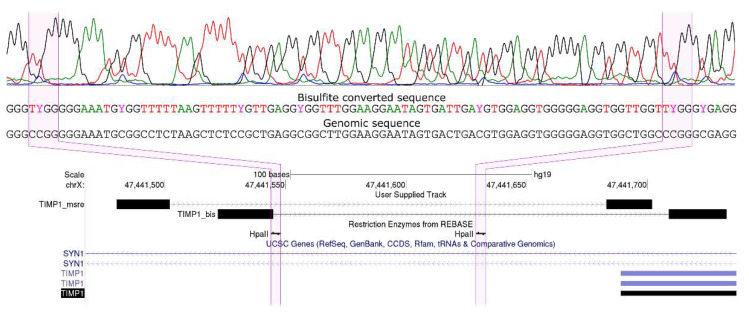
Bisulfite sequence of a fragment of the *TIMP1* gene promoter region, validating partially methylated status of CpGs within the HpaII sites assessed by MSRE-PCR in this study. Blue peaks on the sequence correspond to methylated cytosines. HpaII sites are highlighted. TIMP1_msre, an MSRE-PCR product; primers are black rectangles and the insert is a tiny line. TIMP1_bis, a bisulfite PCR product; primers are black rectangles and the insert is a tiny line.

**Figure 3 biomedicines-08-00116-f003:**
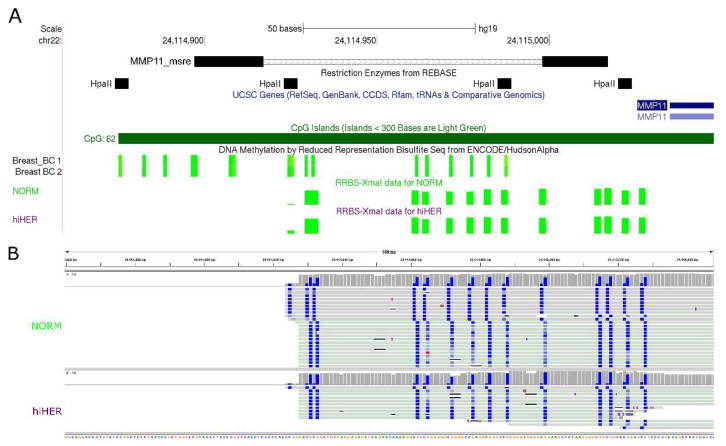
Methylation of a fragment of the *MMP11* gene promoter assessed by MSRE-PCR and reduced representation bisulfite sequencing (RRBS). (**A**) MMP11_msre, an MSRE-PCR product; primers are black rectangles and the insert is a tiny line. Breast_BC1 and Breast_BC2, ENCODE RRBS results: green is for non-methylated CpGs. NORM, XmaI-RRBS results for normal breast tissues. hiHER, XmaI-RRBS results for HER-positive breast tumors of the CpG island hypermethylated subtype. Green is for non-methylated CpGs. (**B**) Examples of clonal bisulfite sequences obtained by by XmaI-RRBS. NORM, one of the normal breast tissue samples. hiHER, one of the HER-positive breast tumors of the CpG island hypermethylated subtype. Blue is for non-methylated CpGs.

**Figure 4 biomedicines-08-00116-f004:**
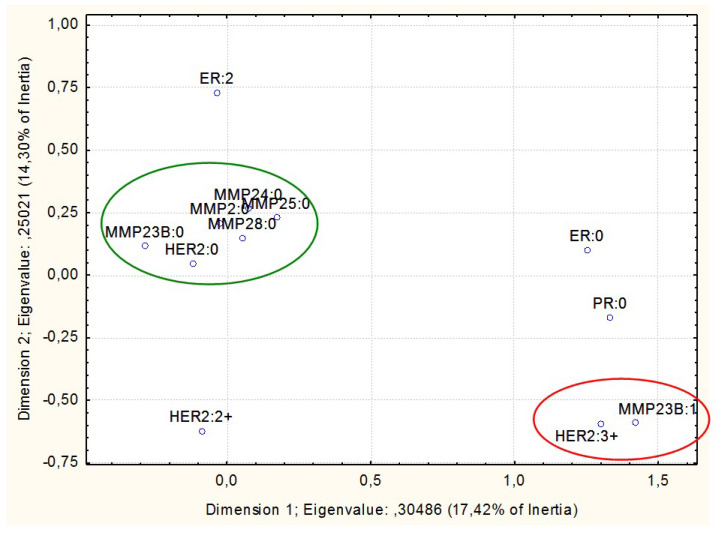
Results of multiple correspondence analysis of HER2 expression status vs. the methylation status of differentially methylated promoter CpG dinucleotides of matrix metalloproteinases genes in breast cancer samples. The non-methylated (designated as “0” at the gene symbol) status of *MMP2, MMP23B, MMP24, MMP25*, and *MMP28* differentially methylated promoter CpG dinucleotides is associated with lack of HER2 expression (green ellipse). The methylated status of *MMP23B* differentially methylated promoter CpG dinucleotides is associated with a high level (3+) of HER2 expression (red ellipse).

**Figure 5 biomedicines-08-00116-f005:**
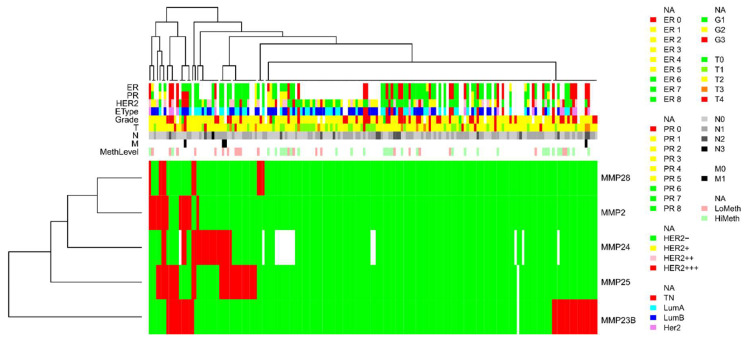
Cluster analysis of the DNA methylation data obtained for 183 breast cancer samples by methylation-sensitive restriction enzyme digestion PCR (MSRE-PCR) of the promoter CpG islands of the *MMP2, MMP23B, MMP24, MMP25,* and *MMP28* genes. LoMeth and HiMeth (MethLevel track) are the moderately methylated and hypermethylated breast cancer (BC) subtypes previously assigned to 100 samples from the same tissue collection on the basis of a genome-wide DNA methylotyping [15]. The expression levels of the estrogen (ER) and progesterone (PR) receptors and the HER2 oncoprotein, breast cancer expression subtype (EType: TN, triple negative; LumA, luminal A; LumB, luminal B; and Her2, HER2 positive), tumor grade (G1–G3) and size (T0–T4), and regional lymph node (N0–N3) and distant (M0–M1) metastasis status are also indicated as colored rectangles along the corresponding tracks. The heat map demonstrates methylated (red) or nonmethylated (green) methylation statuses of the assessed promoter CpG dinucleotides of the matrix metalloproteinases (MMP) genes assayed in breast cancer samples.

**Table 1 biomedicines-08-00116-t001:** Primers used to assess the methylation status of the *MPP* and *TIMP* genes by methylation-sensitive restriction enzyme digestion PCR.

Gene	Primers	PCR Product Size, bp	Number of HpaII Sites within MSRE-PCR Fragment
*MMP2*	F: TACAAAGGGATTGCCAGGAC R: CATTAGCGCCTCCATCGTAG	239	3
*MMP11*	F: GTACCCTCCCCGTTCACCTC R: GCCGCCCCTTATAGCTTCC	120	2
*MMP14*	F: GCCGACAGCGGTCTAGGAAT R: CAGGGGGAGCAGGAGACAAC	180	3
*MMP15*	F: CTCCTCGGGCTTGGGAATTT R: CCAGCTCGGAACACTGCAC	155	4
*MMP16*	F: CGAGAGGCAGCGGCGAAG R: CGGAACCGCCGGTGAACTTA	100	3
*MMP17*	F: CCGGCCTCGTTAGCATACAT R: CCCTCCGCTTCGCGTTCC	125	3
*MMP21*	F: GCCACTCCTCCCTCTCAGC R: CCACCCAGCCCGAGAGTC	250	2
*MMP23B*	F: ACCACACCGGGCTGTAACC R: AGGAGGCACAGGGCGACCA	220	5
*MMP24*	F: CAGAGCCGCTCCTCAGTCTC R: AGGAGGGGGAAGAGGCTAAA	174	2
*MMP25*	F: CTCCCGCGCCCTCTCAAC R: GAAGTGCGCGGTGGAGTC	101	2
*MMP28*	F: CGTGCCTGTGTGGTTCCAG R: CCTGTCAGAACTCGGCAGTC	150	2
*TIMP1*	F: TGAGTCATAGGGAGCTTGGGGG R: CGGGCCGACGAAAGGAGAT	223	2
*TIMP2*	F: AAGCAGCGTCGCCAGCAG R: CCCCCGAGACAAAGAGGAGA	246	3
*TIMP3*	F: CCCTCACCTGTGGAAGCGGT R: CAGACCAATGGCAGAGCCGCA	318	4
*TIMP4*	F: ACCCCCTGCTGTGGACCTC R: CAAGCTGGGTGCTGTTGCTG	150	2
*CUX1*	F: GCCCCCGAGGACGCCGCTACC R: AGGCGGTCCAGGGGTCCAGGC	565	6
*SNRK*	F: GCTGGGTGCGGGGTTTCGGCG R: CGGAGGCTACTGAGGCGGCGG	165	3

**Table 2 biomedicines-08-00116-t002:** Matrix metalloproteinases genes with promoter 5’-cytosine-phosphate-guanine-3’ (CpG) dinucleotides differentially methylated in breast cancer.

Gene	Methylated in Breast Cancer	Methylated in Normal Breast Tissues	Presence (+) or Absence (–) of Methylation in Breast Cancer Cell Lines				
			ZR751	MCF7	T47D	BT474	HS578T
*MMP2*	7.7% (14/183)	0	+	+	+	−	−
*MMP23B*	17% (31/182)	0	+	+	+	−	+
*MMP24*	11.9% (20/168)	0	−	−	-	+	-
*MMP25*	15.4% (28/182)	0	−	+	+	−	−
*MMP28*	4.9% (9/183)	0	+	+	+	+	−

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
