# Peer review of "Abnormal Hypermethylation of CpG Dinucleotides in Promoter Regions of Matrix Metalloproteinases Genes in Breast Cancer and its Relation to Epigenomic Subtypes and HER2 Overexpression"

_biomedicines, 2020, doi:10.3390/biomedicines8050116_

Round 1

Reviewer 1 Report

The manuscript by Simonova et al presents a study that addresses the methylation status of MMP and TIMP family enzymes in breast cancer compared to normal breast tissues. The manuscript addresses an interesting question, but currently suffers of several methodological flaws which need to be addressed before being acceptable for publication.  

The major issue I have is that with the MSRE-PCR that is used only a small subset of methylation sites in the promoter region of the genes can be addressed (namely the ones that harbour a recognition sequence for the restriction enzyme used). Given that promoter regions usually harbour many CpG islands in different sequence contexts it seems exaggerated to extrapolate from the methylation status of one target sequence site to the entire promoter region of the respective gene. Also, I am missing confirmation of at least a subset of the results with another method. In line with this, I don’t quite get this point in the methods: “MSRE-PCR assays were designed with three primer pairs for each locus under study. One fragment was amplified from the gene under study, another one served as a positive PCR control (a constitutively methylated region of the CUX1 gene and a third one was used to check the completeness of DNA hydrolysis…); as it stands, I only see one locus-specific primer pair per target, not three…

A second issue I have is with the type of tissue that was selected for the study: was it fresh-frozen tissue, or FFPE? Nothing is mentioned about this. How much tissue was used per sample? Also, given that the amount of tumor cell to stroma ratio in each sample is bound to influence the results, it would be important to know the approximate amount of stroma that was present in each sample. More ideally, only tumour cells would be analysed, but this is probably not feasible in the presented setup.

Finally, I am missing a careful discussion of the limitations of these points and aspects that I just mentioned.

Finally, there is no mention regarding informed consent in the M&M section – I do hope this was collected before performing the study.

Minor points:

  • Results: it remains unclear to the reader how tissue was sourced, what types of cases were included etc without going into the M&M section. Please add a description of the study setup/methodology at the beginning of this section
  • Abstract and rest of the text: what is ‘abnormal methylation’? Hyper- or Hypo? Compared to what? Rephrase.
  • Abstract: add a conclusive sentence at the end to point out the implications of the study.
  • P2, line 13: “the” families  change to “both protein” families
  • P2, line 26: remove “processes”; also add (EMT), since the term is used in abbreviated form afterwards
  • P2, line 41: change ‘never methylated’ to non-methylated
  • P3, line 8: remove comma after cancer

Author Response

Response to Reviewer 1 Comments

Major points:

Point 1: The major issue I have is that with the MSRE-PCR that is used only a small subset of methylation sites in the promoter region of the genes can be addressed (namely the ones that harbour a recognition sequence for the restriction enzyme used). Given that promoter regions usually harbour many CpG islands in different sequence contexts it seems exaggerated to extrapolate from the methylation status of one target sequence site to the entire promoter region of the respective gene.

Response 1: We agree that MSRE-PCR does not provide enough information that could possibly be extrapolated to the entire promoter region of the respective gene. We now clearly indicate that it was methylation of only a subset of CpGs in the promoter regions of the target genes that we assessed. This is now indicated in the text (lines 31, 35-36, 80, 84, 111-112, 122, 123, 176-178, 204, 351, 361) as well as in the title, captions to Figures 1&2 and Table 1.

We now discuss the limitations of the method in the Discussion (lines 171-185), as follows:

“In this study, we have used methylation-sensitive restriction enzyme digestion PCR (MSRE-PCR) as a principal method to assess DNA methylation of CpG dinucleotides in the promoter regions of 15 target genes in a collection of 378 samples. MSRE-PCR is a very simple and affordable approach to CpG methylation assessment requiring small amounts of DNA, which makes it attractive to use when collections of significant sample size are under study. Yet, MSRE-PCR imposes certain limitations on the results, because only a small subset of methylation sites in the promoter regions of the genes can be addressed, namely the ones that harbour a recognition sequence for the restriction enzyme used. Methylation in these positions doesn't fully reflect the promoter methylation despite the general trend of DNA methylation being correlated within a certain proximity. As far as in our study we used DNA methylation for the purpose of screening for differentially methylated CpGs rather than for comprehensive characterization of DNA methylation along the whole length of the gene promoter sequences, we consider MSRE-PCR method appropriate. The validity of the results obtained in our study is also supported by comparison with RRBS datasets performed either on the same samples or on biologically related samples (normal breast tissue and exhaustively studied BC cell lines)”.

Point 2: Also, I am missing confirmation of at least a subset of the results with another method.

Response 2:

(1) We have added the following to the Results: “MSRE-PCR results were selectively validated by Sanger bisulfite sequencing of the same samples and by comparing MSRE-PCR results for the same samples with two RRBS datasets, one from the ENCODE project [13], and another from our previous XmaI-RRBS study [14] performed on a subset of 111 BC samples and 6 normal breast samples from the collection described here” (lines 95-98).

The text in the next paragraph in Results describes successful validation with cell lines and a normal breast tissue from ENCODE.

(2) We now describe bisulfite sequencing by Sanger in Materials and Methods, lines 315-325, and provide Figure 4 as an illustration of the sequence itself together with the genomic positions of the MSRE-PCR product, the bisulfite PCR product, and the positions of HpaII sites.

(3) To illustrate the validation by comparison with our XmaI-RRBS dataset, and with ENCODE RRBS, we have added Figure 5, and a brief description to the M&M section (lines 332-336).

(4) We have added comparison of our CpG differential methylation results with The Cancer Genome Atlas 450k Infinium chip data for breast cancer (lines 192-200), and conclude that TCGA data for the probes that coincide with, or are in close vicinity to the loci assessed by MSRE-PCR in our study, well support hypermethylation of these regions of the MMP2, MMP23B, MMP24, MMP25, and MMP28 promoters in BC, providing additional validation to our data.

Point 3: In line with this, I don’t quite get this point in the methods: “MSRE-PCR assays were designed with three primer pairs for each locus under study. One fragment was amplified from the gene under study, another one served as a positive PCR control (a constitutively methylated region of the CUX1 gene and a third one was used to check the completeness of DNA hydrolysis…); as it stands, I only see one locus-specific primer pair per target, not three…

Response 3: This phrase was poorly formulated by us. Indeed, there is one locus-specific primer pair per target, and two others in this triplex assay are for the controls. We have rewritten the phrase as follows (lines 289-290):

“Triplex MSRE-PCR assays were used for each locus under study, where one fragment was amplified from the target gene, another one served as a positive PCR control…”

Point 4: A second issue I have is with the type of tissue that was selected for the study: was it fresh-frozen tissue, or FFPE? Nothing is mentioned about this. How much tissue was used per sample? Also, given that the amount of tumor cell to stroma ratio in each sample is bound to influence the results, it would be important to know the approximate amount of stroma that was present in each sample. More ideally, only tumour cells would be analysed, but this is probably not feasible in the presented setup.

Response 4: We now describe these issues in the M&M section,

lines 260-265:

“All BC and matched samples were obtained at surgery of cancer cases none of which underwent neoadjuvant chemotherapy. All tissue samples were fresh-frozen. Fragments of tissues for DNA analysis were examined macro- and micromorphologically. The approximate amount of tumor cells in each sample was estimated to exceed 80%. Fragments of about 10 mg were used for DNA extraction”.

And lines 186-190 in the Discussion:

“As far as MSRE-PCR is rather a qualitative than a quantitative method, we have not performed laser microdissection of the tissue samples prior to DNA extraction. At the same time, MSRE-PCR is very sensitive to methylated alleles. It can identify methylation in a heterogeneous mix containing <2% of cells with methylated fragments [17]. This allows us to use MSRE-PCR for abnormal DNA methylation screening in clinical samples for which fraction of tumor cells is only roughly estimated”.

Point 5: Finally, there is no mention regarding informed consent in the M&M section – I do hope this was collected before performing the study.

Response 5: We have added this information (lines 266-267):

“This study was approved by the Institutional Ethics Committee of the Research Centre for Medical Genetics. Written informed consent was obtained from each participant of this study”.

Minor points:

Point 6: Results: it remains unclear to the reader how tissue was sourced, what types of cases were included etc without going into the M&M section. Please add a description of the study setup/methodology at the beginning of this section.

Response 6: Done (lines 80-83).

Point 7: Abstract and rest of the text: what is ‘abnormal methylation’? Hyper- or Hypo? Compared to what? Rephrase.

Response 7: All the CpGs that were found to be differentially methylated in our study appeared to be non-methylated in normal breast tissues, thus abnormal methylation detected in cancer samples was hypermethylation. We now indicate this where appropriate in the text (lines 35, 38, 41, 87, 113, 201, 204, 347, 351, 352, 361), as well as in the title.

Point 8: Abstract: add a conclusive sentence at the end to point out the implications of the study.

Response 8: Done (lines 41-44).

Point 9: P2, line 13: “the” families change to “both protein” families.

Response 9: Done (line 58).

Point 10: P2, line 26: remove “processes”; also add (EMT), since the term is used in abbreviated form afterwards.

Response 10: Done (line 71).

Point 11: P2, line 41: change ‘never methylated’ to non-methylated.

Response 11: Done (line 90).

Point 12: P3, line 8: remove comma after cancer.

Response 12: Done (line 113).

Reviewer 2 Report

Authors present evidence of cancer-specific changes to DNA methylation in promoters of genes involved in maintaining tissue structure, such as matrix metalloproteinases (MMPs) and their tissue inhibitors (TIMPs).

Although the results seem solid, this manuscript can be further improved:

  1. Authors report that the expression levels of the studied genes have already been known to be associated with cancer. Is anything known about DNA methylation changes in these promoters? Have they been tested in any other cancer study (maybe on a different tissue) using another method?
  2. Methylation sensitive restriction enzyme digestion PCR (MSRE-PCR) is a limited and tricky method. It allows estimating methylation at 1 to several CpG positions per region. The authors used HpaII restriction enzyme. How many restriction sites were there for every promoter region? How were the regions with multiple HpaII restriction sites dealt with?
  3. Since MSRE-PCR provides such limited information, it doesn't seem correct to refer to "methylated genes". First of all, methylation in this position (or a few positions within a small region) doesn't fully reflect the promoter methylation despite the general trend of DNA methylation being correlated within a certain proximity. Second, DNA methylation can also occur within the gene body.
  4. Figure 2 has such a low resolution that it is hard to read the text on it

Minor issues:

1. The phrase "abnormal methylation" is confusing because it doesn't show the sign of the effect. 

2. It is unclear whether authors evaluated expression levels of the genes of interest (mainly HER2) or if this information was obtained from another study.

3. Some phrases can be slightly changed:

 (4.2) "Of notice is the cluster of samples" -> a remarkable cluster... 

(4.28) "a hypothesis may be put forward" -> we hypothesize

Author Response

Response to Reviewer 2 Comments

Major issues:

Point 1: Authors report that the expression levels of the studied genes have already been known to be associated with cancer. Is anything known about DNA methylation changes in these promoters? Have they been tested in any other cancer study (maybe on a different tissue) using another method?

Response 1: We now address this in the Discussion (lines 192-200):

“By now, the most comprehensive dataset on DNA methylation in BC is provided by The Cancer Genome Atlas (TCGA). We gained access to TCGA data through an interactive viewer Wanderer [18], to explore MMPs DNA methylation in breast cancer, with level 3 TCGA data for methylation arrays (450k Infinium chip). It should be borne in mind, that 450k Infinium chip probes do not cover each CpG dinucleotide in the genes promoter CpG islands. TCGA data for the probes that coincide with, or are in close vicinity to the loci assessed by MSRE-PCR in our study, well support hypermethylation of these regions of the MMP2, MMP23B, MMP24, MMP25, and MMP28 promoters in BC, providing additional validation to our data.

Point 2: Methylation sensitive restriction enzyme digestion PCR (MSRE-PCR) is a limited and tricky method. It allows estimating methylation at 1 to several CpG positions per region. The authors used HpaII restriction enzyme. How many restriction sites were there for every promoter region? How were the regions with multiple HpaII restriction sites dealt with?

Response 2: We have supplemented Table 2 with a column indicating numbers of HpaII sites within MSRE-PCR fragment. To address the question of how the regions with multiple HpaII restriction sites were dealt with, we added the following to the gel description in Figure 3 caption (lines 311-314):

“MSRE-PCR does not provide information on the methylation status of individual CpGs contained within the restriction enzyme recognition sequence in an assessed locus. Thus, positive MSRE-PCR signal was interpreted as hypermethylation of the whole target locus, while negative MSRE-PCR signal, as its non-methylated state”.

Point 3: Since MSRE-PCR provides such limited information, it doesn't seem correct to refer to "methylated genes". First of all, methylation in this position (or a few positions within a small region) doesn't fully reflect the promoter methylation despite the general trend of DNA methylation being correlated within a certain proximity. Second, DNA methylation can also occur within the gene body.

Response 3: We agree that MSRE-PCR does not provide enough information to speak of "methylated genes". We now clearly indicate that it was methylation of a subset of CpGs in the promoter regions of the target genes that we assessed. This is now indicated in the text (lines 31, 35-36, 80, 84, 111-112, 122, 123, 176-178, 204, 351, 361) as well as in the title, captions to Figures 1&2 and Table 1.

We now discuss the limitations of the method in the Discussion (lines 171-185), as follows:

“In this study, we have used methylation-sensitive restriction enzyme digestion PCR (MSRE-PCR) as a principal method to assess DNA methylation of CpG dinucleotides in the promoter regions of 15 target genes in a collection of 378 samples. MSRE-PCR is a very simple and affordable approach to CpG methylation assessment requiring small amounts of DNA, which makes it attractive to use when collections of significant sample size are under study. Yet, MSRE-PCR imposes certain limitations on the results, because only a small subset of methylation sites in the promoter regions of the genes can be addressed, namely the ones that harbour a recognition sequence for the restriction enzyme used. Methylation in these positions doesn't fully reflect the promoter methylation despite the general trend of DNA methylation being correlated within a certain proximity. As far as in our study we used DNA methylation for the purpose of screening for differentially methylated CpGs rather than for comprehensive characterization of DNA methylation along the whole length of the gene promoter sequences, we consider MSRE-PCR method appropriate. The validity of the results obtained in our study is also supported by comparison with RRBS datasets performed either on the same samples or on biologically related samples (normal breast tissue and exhaustively studied BC cell lines)”.

Point 4: Figure 2 has such a low resolution that it is hard to read the text on it.

Response 4: Figure 4 has high resolution, but there is not enough space for it in portrait orientation to be clearly read. We now provide it in landscape orientation.

Minor issues:

Point 5: The phrase "abnormal methylation" is confusing because it doesn't show the sign of the effect.

Response 5: All the CpGs that were found to be differentially methylated in our study appeared to be non-methylated in normal breast tissues, thus abnormal methylation detected in cancer samples was hypermethylation. We now indicate this where appropriate in the text (lines 35, 38, 41, 87, 113, 201, 204, 347, 351, 352, 361), as well as in the title.

Point 6: It is unclear whether authors evaluated expression levels of the genes of interest (mainly HER2) or if this information was obtained from another study.

Response 6: For HER2, as well as for ER, PR it was not mRNA expression, but protein expression assessed by immunohistochemistry, thus we provide HER2 scores as ihc scores. We have supplemented Materials and Methods with the following (lines 263-264):

“expression of ER, PR and HER2 receptors was evaluated by immunohistochemistry”

Point 7: Some phrases can be slightly changed:

 (4.2) "Of notice is the cluster of samples" -> a remarkable cluster...

 (4.28) "a hypothesis may be put forward" -> we hypothesize

Response 7: Done (lines 122 & 148).

Round 2

Reviewer 1 Report

The authors have addressed all the concerns in a satisfactory way. Some english editing of the text might be beneficial.